# A Lipid Bilayer Formed on a Hydrogel Bead for Single Ion Channel Recordings

**DOI:** 10.3390/mi11121070

**Published:** 2020-12-01

**Authors:** Minako Hirano, Daiki Yamamoto, Mami Asakura, Tohru Hayakawa, Shintaro Mise, Akinobu Matsumoto, Toru Ide

**Affiliations:** 1Bio Photonics Laboratory, The Graduate School for the Creation of New Photonics Industries, Shizuoka 431-1202, Japan; hirano37@gpi.ac.jp; 2Graduate School of Interdisciplinary Science and Engineering in Health Systems, Okayama University, Okayama 700-8530, Japan; pluy0evf@s.okayama-u.ac.jp (D.Y.); asakura@okayama-u.ac.jp (M.A.); hayaka-t@cc.okayama-u.ac.jp (T.H.); 3Department of Molecular and Cellular Biology, Medical Institute of Bioregulation, Kyushu University, Fukuoka 812-8582, Japan; s_mise@bioreg.kyushu-u.ac.jp (S.M.); akinobu@bioreg.kyushu-u.ac.jp (A.M.)

**Keywords:** ion channel, lipid bilayer, single-channel recording

## Abstract

Ion channel proteins play important roles in various cell functions, making them attractive drug targets. Artificial lipid bilayer recording is a technique used to measure the ion transport activities of channel proteins with high sensitivity and accuracy. However, the measurement efficiency is low. In order to improve the efficiency, we developed a method that allows us to form bilayers on a hydrogel bead and record channel currents promptly. We tested our system by measuring the activities of various types of channels, including gramicidin, alamethicin, α-hemolysin, a voltage-dependent anion channel 1 (VDAC1), a voltage- and calcium-activated large conductance potassium channel (BK channel), and a potassium channel from *Streptomyces lividans* (KcsA channel). We confirmed the ability for enhanced measurement efficiency and measurement system miniaturizion.

## 1. Introduction

Ion channel proteins are membrane proteins that regulate various biological functions by controlling ion flow across cell membranes in response to stimuli [1,2]. Ion channel proteins play important roles in a wide variety of cell functions, making them important drug targets [3,4]. For example, in nerve cells, the influx of Na^+^ ions thorough voltage-gated Na^+^ channels produces a nerve signal [1,2], while voltage-gated calcium channels regulate gene expressions [5]. At the same time, the dysfunctional activity of ion channels can cause serious and even fatal diseases; the activity of the P2X purinoceptor 4 (P2X4) channel is related to neuropathic pain [6,7], malfunction of the cystic fibrosis transmembrane conductance regulator (CFTR) channel causes cystic fibrosis [8,9], and disorder of the human ether-a-go-go related gene (HERG) channel leads to serious arrhythmia [10].

The artificial lipid bilayer recording technique measures the ion transport activities of channel proteins along with other ion transporting proteins and peptides [11]. This technique allows researchers to determine the biophysical and pharmacological properties of the channel with high sensitivity and accuracy. However, the measurement efficiency of the technique is low. In fact, more than several minutes and sometimes tens of minutes are required to measure ion currents due to the time-consuming process of making artificial lipid bilayer membranes and the low incorporation rate of channels into the membranes [11,12]. 

The physical processes that occur during lipid bilayer formation have been extensively studied and are described in standard textbooks [11]. Principally, the lipid bilayer membrane forms when organic solvent and inverted lipid micelles are removed between the two lipid monolayers. Bilayers cannot be made from a lipid solution without contacting the two monolayers. In the most frequently used bilayer forming technique, i.e., the painting technique, a lipid solution is painted over a small hole in a thin plastic sheet under the aqueous solution and the organic solvent is spontaneously removed between the two lipid monolayers. The removal takes several tens of minutes at most.

Membrane proteins, including channel proteins, are generally not inserted into bilayers directly from the aqueous solution, with certain exceptions such as voltage-dependent anion channels (VDACs). The most efficient method used to incorporate channel proteins into bilayers is vesicle fusion. However, because vesicles stochastically fuse with the bilayer, the process cannot be controlled, and thus sometimes takes a long time.

In order to improve the measurement efficiency, several approaches have been proposed. For example, bilayer membranes can be promptly made in a lipid solution at the droplet–droplet interface [13,14,15], droplet–aqueous solution interface [16], droplet–agarose gel interface [14], or agarose gel–agarose gel interface [17]. In addition, in order to enhance the incorporation rate of the channel proteins into artificial membranes, we developed a technique whereby the incorporation happens simultaneously with the formation of the membrane on a supporting substrate [18,19]. We reported a method that measures the ion current of potassium channels from *Streptomyces lividans* (KcsA) or from *Methanobacterium thermoautotrophicum* (MthK) channels by binding them to cobalt affinity gel beads with a histidine tag and then forming a lipid bilayer membrane with the channels on the bead by pushing the lipid layer onto the bead [18]. In another study, we used a gold electrode covered with polyethylene glycol (PEG), on which the ion channels were immobilized [19]. By contacting the lipid monolayer at the surface of the probe with another lipid monolayer formed at the interface between the lipid solution and aqueous recording solution, a lipid bilayer membrane was formed at the same time as the channels were incorporated into the membrane. Since lipid bilayer membranes containing ion channels were formed in a single step, channel currents could be measured with high efficiency [19,20,21]. 

In general, we have developed techniques for channel current recordings using various types of substrate-supported bilayers to improve the measurement efficiency of conventional artificial bilayer methods. Here, we report further enhanced efficiency by simplifying the bilayer formation technique and shortening the required time for the measurement preparation. This component technology could potentially lead to the miniaturization of a measurement apparatus. 

## 2. Materials and Methods 

### 2.1. Materials

Asolectin, alamethicin, and gramicidin were purchased from Sigma (St. Louis, MO, USA), α-hemolysin was from Abcam plc (Cambridge, UK), Co^2+^ affinity gel beads (TALON Metal Affinity Resins) from Clontech (Mountain View, CA, USA), and Sepharose 4B beads were from Cytiva Japan (Tokyo, Japan). All other chemicals were commercial products of analytical grade.

### 2.2. Constructs and Mutants 

The cDNA of mouse voltage-dependent anion channel 1 (VDAC1) (ENSMUST00000102758.7) was subcloned into a multi-cloning site (NdeI-XhoI) of the pET21b vector. The potassium channels from *Streptomyces lividans* (KcsA channel) gene cloned into pQE-30 vectors, including an N-terminal hexahistidine tag, was a gift from Dr. Kubo of the National Institute of Advanced Industrial Science and Technology. We used a KcsA mutant (E71A) made by using the QuickChangeTM site-directed mutagenesis kit (Stratagene), because the E71A mutation is known to inhibit inactivation.

### 2.3. Protein Expression and Purification of Voltage-Dependent Anion Channel 1 (VDAC1)

The pET21b plasmids containing the VDAC1 sequence were transformed into *Escherichia coli* BL21(DE3) and then overexpressed in the presence of 1 mM isopropyl-d-thiogalactopyranoside (IPTG) for 4 h. The BL21(DE3) pellets were incubated in BugBuster Protein Extraction Reagent (Merck, Kenilworth, NJ, USA) containing protease inhibitors (cOmplete™ Protease Inhibitor Cocktail EDTA-free, Roche, Basel, Switzerland) and benzonase (Merck) for 5 min at room temperature, and then incubated in BugBuster Protein Extraction Reagent containing 250 μg/mL lysozyme (Merck) for 5 min at room temperature. After they were centrifuged at 16,000× *g* for 20 min, the precipitates were sonicated in binding buffer (6 M GHCl, 300 mM NaCl, 10 mM imidazole, 50 mM Tris-HCl (pH 8.0)) and then centrifuged at 16,000× *g* for 10 min. The supernatant was mixed with nickel resin (Thermo Fisher, Waltham, MA, USA) and rotated at 4 °C for 60 min. Nonspecific bound proteins were removed with 5-fold volume of binding buffer, 5-fold volume of wash buffer (6 M GHCl, 300 mM NaCl, 20 mM imidazole, 50 mM Tris-HCl (pH 8.0)), and 5-fold volume of refolding buffer (100 mM NaCl, 0.4% *n*-dodecyl-*N*,*N*-Dimethylamine-*N*-oxide (LDAO, Affymetrix), 50 mM Tris-HCl (pH 8.0)). Then, VDAC1 was eluted with elution buffer (100 mM NaCl, 0.4% LDAO, 500 mM imidazole, 50 mM Tris-HCl (pH 8.0)) and dialyzed with a dialysis membrane (Slide-A-Lyzer MINI Dialysis Device, 10K MVCO, Thermo Fisher). VDAC1 protein was further purified by cation exchange using ÄKTA explorer 10S (GE Healthcare, Chicago, IL, USA) and RESOURCE™ S column (GE Healthcare).

### 2.4. Protein Expression and Purification of KcsA

The pQE-30 plasmids containing the KcsA sequence were transformed into *Escherichia coli* XL1-Blue and overexpressed by the addition of IPTG to a final concentration of 0.5 mM for 2 h. The expressed channels were extracted from membrane fractions by 10 mM *n*-dodecyl-d-maltoside (DM, Dojin, Kumamoto, Japan). Co^2+^ affinity gel beads (TALON Metal Affinity Resins, Clontech) equilibrated with normal buffer (5 mM DM, 20 mM Tris-HCl (pH7.6), 100 mM KCl) were added to the extracted channel protein solution and incubated for 30 min at 4 °C in order to bind histidine tags to the gel beads. Nonspecific bound proteins were removed with washing buffer (normal buffer containing 20 mM imidazol). 

### 2.5. Isolation of Plasma Membrane Vesicles from Myometrium

Myometrium plasma membrane vesicles containing BK channels (a voltage- and calcium-activated large conductance potassium channel) were prepared according to the method of Toro et al. [22]. Membrane vesicles were isolated from pig uteri. Connective tissue and endometrium were removed from the uteri. The tissue was homogenized in sucrose solution (300 mM sucrose and 20 mM Tris-HEPES, pH 7.2) in the presence of protease inhibitors. The homogenate was centrifuged for 30 min at 700× *g*. The supernatant was centrifuged for 40 min at 14,000× *g* and then for 20 min at 9500× *g*. The pellet was resuspended in solution (600 mM KC1 and 5 mM Na-PIPES, pH 6.8) with a Teflon pestle homogenizer. Samples were incubated on ice for 1 h and centrifuged for 1 h at 42,000× *g*. The pellets were resuspended in 400 mM KCl, 10% sucrose (*w*/*w*), and 5 mM Na-PIPES, pH 6.8. The microsomes were placed on top of a discontinuous sucrose gradient (*w*/*w*): 20:25:30:35:40%. The gradient was centrifuged at 67,500× *g* in a swinging rotor for 18 h. Membranes obtained from the 20:25% sucrose interface were used in this study.

### 2.6. Formation of Bilayers on a Gel Bead and Current Recordings 1

A hydrogel bead was immobilized to the aperture at the glass pipette tip by suction. A Co^2+^ affinity gel bead, on which KcsA proteins were immobilized, was used in the case of KcsA channel recordings and a Sepharose 4B bead was used for other types of channels. A glass capillary (GC150T-10, Harvard Apparatus, Holliston, MA, USA) was pulled to make a pipette with a very fine tip (<1 μm) using a pipette puller (P-97, Sutter Instrument, Novato, CA, USA), then the tip was cut to make a large aperture (>10 μm). The tip was fabricated using a microforge (MF-900, Narishige, Tokyo, Japan) in order to smooth the surface. Figure 1A shows a Co^2+^ affinity gel bead fixed at the tip of a glass pipette by suction. The glass pipette was set with a pipette holder and a bead was fixed at the tip of the pipette by suctioning it through a tube from the holder with a syringe. In Figure 1A, the bead is approximately 80 μm in diameter and the pipette aperture is 30 μm in diameter. Prior to the electrical recording, the beads were incubated in order to be equilibrated with the recording solution for 30 min. 

Figure 1B shows a schematic drawing of the experimental apparatus (see also Figure A1 and Figure A2). A cut glass tube (internal diameter, 6 mm) glued to a slide glass was used as the bath chamber. An aqueous recording solution was poured into the bath chamber and lipid solution (30 mg/ml asolectin in *n*-decane or hexadecane) was layered over the recording solution layer. Bilayers were made by moving the bead fixed at the pipette tip through this lipid solution layer into the recording solution (Figure 1C). The bead was moved downward until gently contacting the lipid/aqueous solution interface by using a manipulator. Contact between the bead and the interface could be easily seen with a microscope. The depth at which the bead was dipped into the aueous solution was usually 10–100 μm. However, it was not necessary to be precise, such that using a coarse manipulator to move the bead was sufficient. A lipid bilayer membrane was spontaneously formed immediately after the contact. 

One of two channel forming peptides (gramicidin and alamethicin), the hemolytic protein α-hemolysin, or detergent-solubilized VDAC1 was added to the recording solution at appropriate concentrations before the bilayer formation. These molecules were spontaneously incorporated into the bilayer formed on the Sepharose 4B bead to make ionic channels. KcsA channel proteins were spontaneously incorporated into the bilayers by forming bilayers with a Co^2+^ affinity gel bead, on which the channel proteins were immobilized through a histidine tag. BK channels were incorporated into the bilayers by vesicle fusion. Prior to the bilayer formation, a Sepharose 4B immobilized at the glass pipette tip was dipped into a membrane vesicle suspension that contained BK channels. The aqueous recording solution was held at virtual ground level so that the voltage at the solution level in the pipette connected to a patch clamp amplifier (CEZ-2400, Nihonkohden, Tokyo, Japan) by an Ag–AgCl electrode defined the membrane potential. Unless otherwise noted, the signal was filtered at 1 kHz, sampled at 10 kHz, digitized, then stored on a PC. 

### 2.7. Formation of Bilayers on a Gel Bead and Current Recordings 2

As shown in Figure 2, a Sepharose 4B bead was fixed by suction at the tip of a glass pipette in a recording solution. Then, the chamber was filled with a lipid solution (30 mg/mL asolectin in *n*-decane). The aqueous recording solution was extruded out from a polyethylene tube (0.1–0.5 mm inner diameter) placed near the bead by manually exerting positive pressure with a syringe, such that it contacted the bead in the lipid solution. The contact was observed with a microscope. A bilayer membrane was promptly formed at the water–gel interface. The solution in the tube was held at virtual ground level so that the voltage at the solution level in the glass pipette connected to a patch clamp amplifier by an Ag–AgCl electrode defined the membrane potential (see also Figure A3).

## 3. Results and Discussion

### 3.1. Formation of Bilayers on a Gel Bead

The bilayer formation technique developed in this study allowed us to make artificial bilayers for channel recordings much more quickly than with conventional techniques, significantly increasing the measurement efficiency. Figure 1C shows the formation process for the artificial bilayers on a gel bead. As described in the Materials and Methods section, bilayers were spontaneously formed on a gel bead by inserting the bead into an aqueous solution from the lipid solution layer. Figure 1A shows a micrograph of a hydrogel bead fixed at the tip of a glass pipette by suction. In order to prevent the lipid solution from flowing inside the pipette through a gap between the bead and the pipette surface, the tip of the pipette was melted so as to be smoothed by a microforge apparatus for patch clamp pipettes. Even slight roughness on the pipette surface complicated the current measurement, particularly when using a Co^2+^ affinity gel bead, which is a cross-linked agarose bead.

We measured gramicidin channel currents to confirm if the membrane formed at the gel–water interface was a lipid bilayer. In the bilayer formation process, two lipid monolayers, one at the gel–lipid interface and the other at the lipid–water interface, were folded together to make a bilayer. This means organic solvent and lipid reversed micelles should be removed between the two lipid monolayers to make the bilayer. We easily observed contact between the bead and the lipid–water interface using a microscope. However, we could not confirm if a bilayer was formed, even if the bead was in contact with the interface. Therefore, we measured the gramicidin current, because gramicidin must make a tandem dimer in a membrane in order to show current activity, which can be measured only when the membrane becomes thin enough to form a bilayer [23]. 

Figure 3 shows repeated bilayer formations, which were formed by plunging the gel bead into a lipid solution. The current was recorded by moving the gel bead up and down near the lipid–water interface. As shown in the figure, the bead was moved using a piezo-micromanipulator with an amplitude of 10 μm at approximately 1 Hz. The aqueous recording solution contained 250 mM KCl, 5 mM Tris-HCl pH 7.4, and 100 nM gramicidin. As the bead moved downward, the current did not increase until just after the bead reached the interface between the lipid and aqueous solutions, which marked the formation of the bilayer membrane and incorporation of gramicidin. On the other hand, when the bead was moved upward, the current decreased back to zero, indicating that the membrane became thick as lipids and organic solvent flowed into the space between the two lipid monolayers. We incorporated gramicidin into the bilayers by adding it to the recording solution.

Figure 3 demonstrates that we were able to quickly repeat current measurements using the bead. This ability can increase the throughput of pharmacological measurements. As shown in the figure, a bilayer was formed almost instantly after bringing the bead into contact with the aqueous solution. The greatest advantage of this method is that we were able to form artificial bilayers much more rapidly than with conventional methods. In conventional artificial bilayer methods, it takes several minutes or longer to form bilayers [24]. Furthermore, in conventional artificial bilayer methods, extreme care must be taken not to rupture the bilayer when the recording solutions are exchanged. Perfusion often results in a ruptured bilayer membrane, thus requiring the measurement to be recommenced. With the present technique, however, ionic current measurement can be restarted using the same bead within a few seconds, saving much time and labor.

### 3.2. Spontaneous Incorporation of Channels into Bilayers and Current Recordings

The bilayer formation and insertion processes for the channel-forming proteins and peptides were always reproducible. As shown in Figure 3, the bilayers could be repeatedly formed any number of times, while the channel-forming peptides and proteins were invariably inserted into the bilayers from the aqueous solution.

#### 3.2.1. Gramicidin

Figure 4A is a typical current trace of a gramicidin channel. The membrane voltage was held at 50 mV, and the aqueous solution contained 150 mM NaCl, 100 nM gramicidin, and 10 mM MOPS-Tris, pH 7.4. Each step in the current recording corresponds to the unitary current of the gramicidin channel. The single-channel conductance was determined as 10 ± 0.1 pS (*n* = 5), which agreed well with the value obtained by conventional methods [25].

#### 3.2.2. Alamethicin

Figure 4B represents a current recording for alamethicin channels. Alamethicin is a channel-forming peptide antibiotic produced by fungi. It is known to form voltage-dependent ion channels by aggregating four to six molecules in bilayers. The current trace in the figure was taken with a solution containing 1 KCl and 500 nM alamethicin. The single-channel conductance calculated from the I–V relationship was 1.1 ± 0.11 nS (*n* = 5). The current fluctuation showed voltage dependency, as reported in previous papers [26]. The open probability of the channel changed from 0.4 at +100 mV to <0.01 at −100 mV. The properties of the single-channel conductance and voltage dependency observed in this study agreed with conventional experiments, validating our novel technique.

#### 3.2.3. α-Hemolysin

Figure 4C shows the α-hemolysin channel current recorded with our method. α-Hemolysin is a hemolytic protein secreted by *Staphylococcus aureus*, which is known to form a heptamer in bilayers, forming a large ion channel pore [27,28]. The trace in Figure 4C was recorded with a solution containing 100 mM KCl and 10 mM Hepes-Tris, pH 7.3, in the presence of 1 μM α-hemolysin. From the single-channel I–V relationship, the single-channel conductance was calculated to be 245 ± 16 pS (*n* = 5), which was identical to the values obtained by conventional bilayer methods [29], again validating our method. Our technique could potentially be used to measure other types of channel-forming proteins, such as γ-hemolysin and Cry toxins. 

#### 3.2.4. Voltage-Dependent Anion Channel 1 (VDAC1)

Figure 4D shows a current tracing of mouse VDAC 1 taken with 1 M KCl and 10 mM MOPS-Tris, pH 7.4, and 1 μg/mL VDAC1. VDAC1 solubilized with detergent is known to be directly incorporated into artificial bilayers from an aqueous solution [30]. From the slope of the I–V relationship of single-channel currents, the single-channel conductance was calculated to be 1.4 nS (*n* = 5), and the channel showed voltage dependent gating as reported previously [30]. 

The above confirmed that our method can be used to measure the currents of several types of channels. Importantly, the channel current properties observed were in good agreement with the results obtained from other techniques, such as the conventional planar bilayer technique and patch clamp analysis. Thus, our method can be used to study a wide range of channels. 

Moreover, the channel current measurements were relatively easy, as we only needed to form bilayers with recording solutions containing the channel peptides or proteins. All of the channels shown in Figure 4, except for VDAC1, were channel-forming peptides or pore-forming proteins, which spontaneously incorporate into bilayer membranes from an aqueous solution to make ion channel pores. Detergent-solubilized VDAC1 proteins can be easily reconstituted into artificial bilayers from aqueous solution, but most other biological channel proteins cannot be reconstituted in this way.

### 3.3. Incorporation of Channel Proteins Immobilized on a Gel Bead

Next, we immobilized physiological channel proteins (which cannot be directly inserted into bilayers from an aqueous solution) on the surface of a hydrogel bead and reconstituted them into bilayers to measure their currents. Figure 5 shows the results for a single KcsA channel. KcsA is a potassium-selective channel from the soil bacteria *Streptomyces lividans*. It is one of the best-characterized channel proteins because its structure is very simple. As described in the Materials and Methods section, purified KcsA channels were solubilized with detergent and immobilized on a gel bead via histidine tags. The bead was moved to contact the aqueous solution through the lipid solution layer, and immobilized KcsA channels were reconstituted immediately after the bilayer formation.

Figure 5A illustrates the incorporation of the KcsA channel immobilized on a gel bead in a bilayer. Figure 5B shows the current fluctuation of the KcsA channel recorded in a solution containing 200 mM KCl and 20 mM succinate-KOH, pH 4.0. The membrane potential was held at 50 mV. Figure 5C shows the I–V relation of the unitary current. From this relationship, the single-channel conductance was determined to be 200 pS, which agreed well with values obtained using the conventional planar bilayer method [31,32,33].

As we previously reported, some types of physiological channel proteins that have been solubilized with detergent and immobilized on a gel [18] or solid [19] substrate have been successfully reconstituted directly into artificial bilayers for single-channel current measurements. In previous studies, we incorporated channel proteins into bilayers without any additional processes, such as vesicle fusion. Here, we further improved this method by simplifying the measurement to enhance the measurement efficiency. Of note, one limitation was the manual movement of the bead. Automating this step is expected to further improve the efficiency. Furthermore, our system anchored the KcsA channel proteins to the gel at a protein terminal; however, the N- and C-terminals regulate the channel activities [31,34].

### 3.4. Channel Inorporation through Vesicle Fusion

Physiological channel proteins were also measured by incorporating them into bilayers through vesicle fusion. Figure 6 shows the results of single-channel recordings of the BK channel, which is a voltage- and calcium-dependent K^+^ channel purified from pig uterus. Prior to the bilayer formation, a hydrogel bead immobilized at the glass pipette tip was dipped into a membrane vesicle suspension that contained BK channels in the vesicular membrane. Immediately after contact between the gel and lipid–water interface, the current began to fluctuate, showing that BK channels were inserted into the bilayer by vesicle fusion. It is likely that the incorporation rate was high compared with conventional bilayer methods, because the space between the gel and the lipid was narrow, meaning that the vesicle concentration was kept high. In Figure 6A, we show the current traces recorded in a solution containing 250 mM KCl, 5 mM Tris-Cl, pH 7.4. The bilayer was held at the indicated voltages. Figure 6B shows histograms of the current amplitudes at membrane potentials of +100 mV and −100 mV. The open probability changed from >0.9 at +100 mV to <0.1 at −100 mV. These results are consistent with the typical voltage dependency of the BK channel [22]. The single-channel conductance of the BK channel was determined to be 230 pS, agreeing well with the values determined by conventional methods [35].

### 3.5. Simultaneous Measurements of Multiple Channel Types

Bilayers were promptly made by pushing out the aqueous solution from a fine tube, such that the solution contacted a gel bead. The effect of this strategy is to miniaturize the apparatus. Above, we showed that we could measure the properties of various types of channels by using bilayers formed on a gel bead. Although our technique makes the recordings more efficient than the conventional bilayer technique, we should develop a simultaneous parallel recording technique to further increase the measurement efficiency. However, because our apparatus contains a bulky manipulator, it is not suitable for the parallel recordings of many channels. Therefore, we modified the bilayer formation technique to miniaturize the apparatus.

Figure 7A shows a gramicidin current trace recorded with a bilayer made on the surfaces of gel beads by moving the interface between the aqueous and lipid solutions. As illustrated in Figure 2, the aqueous solution was extruded so as to come into contact with the gel bead, thus forming a bilayer at the interface. Immediately after the contact, the current began to fluctuate, showing bilayer formation and the insertion of gramicidin into it. The measured properties were typical of a gramicidin channel. The bilayer was durable against the membrane voltage and stable at ± 200 mV.

Next, we used this modified bilayer formation technology to perform simultaneous recordings of different types of channels. The current recordings shown in Figure 7B are of simultaneously recorded gramicidin and alamethicin. We had previously reported simultaneous recordings of gramicidin channels [17]. In that experimental system, there was a single chamber for the aqueous recording solution, therefore the recording solutions on one side of all of the bilayers were electrically shunted. In contrast, in the present study, each bilayer was made in a different recoding solution, meaning that we could simultaneously measure different types of channels in different environments.

## 4. Conclusions

By improving our previously developed channel recording techniques, here we developed an efficient method for measuring ion channel activities. We measured the natural properties of various types of channels incorporated into gel-supported bilayers. We were able to simultaneously measure different types of channels in different environments. This method allows miniaturization of a recording system. In the future, by making this system multi-channel, we may obtain a high-throughput device for channel current measurement.

## Figures and Tables

**Figure 1 micromachines-11-01070-f001:**
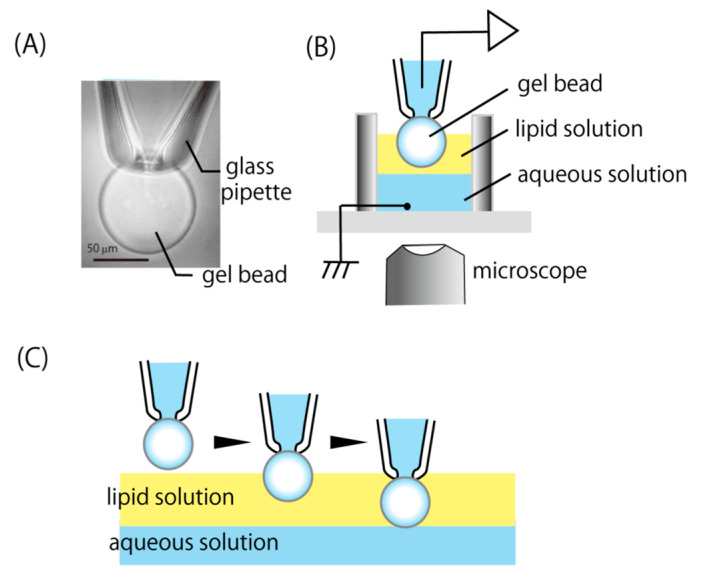
Bilayer formation on a hydrogel bead. (**A**) A micrograph of a Co^2+^ affinity gel bead fixed at the pipette tip by suction. (**B**) The experimental apparatus. The pipette solution and the recording solution were connected with an Ag–AgCl wire to the patch clamp amplifier and virtual ground level, respectively. (**C**) The bilayer formation process. A gel bead fixed at the tip of a glass pipette was plunged into a lipid solution layer and inserted into an aqueous recording solution through the lipid solution layer. The bilayer is spontaneously formed at the gel–water interface.

**Figure 2 micromachines-11-01070-f002:**
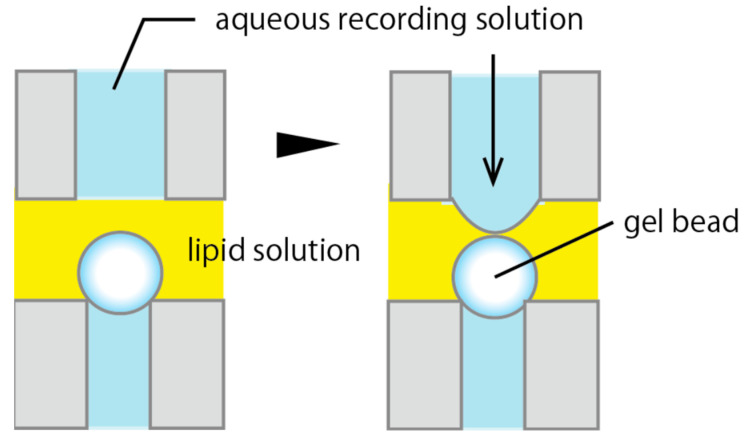
A bilayer was formed at the water–gel interface by pushing the recording solution out from a tube.

**Figure 3 micromachines-11-01070-f003:**
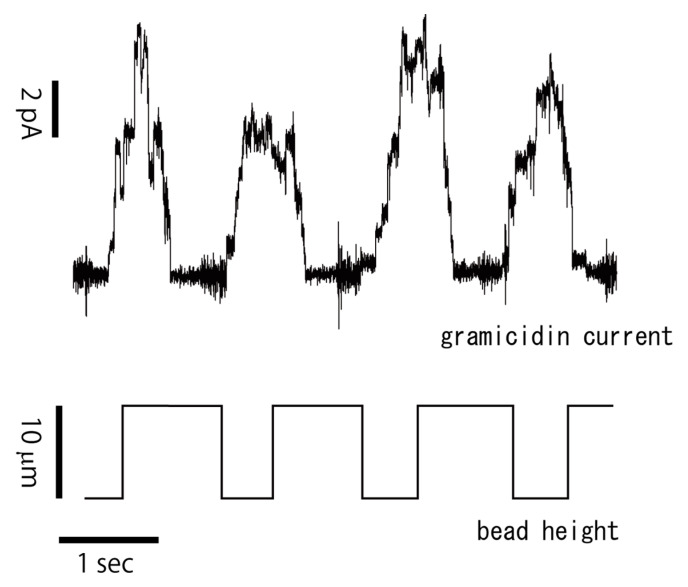
Repeated formation of bilayers on a gel bead. The bead was moved up and down near the water–lipid interface in the presence of gramicidin in the recording solution. The height of the bead was changed by 10 μm at approximately 1 Hz.

**Figure 4 micromachines-11-01070-f004:**
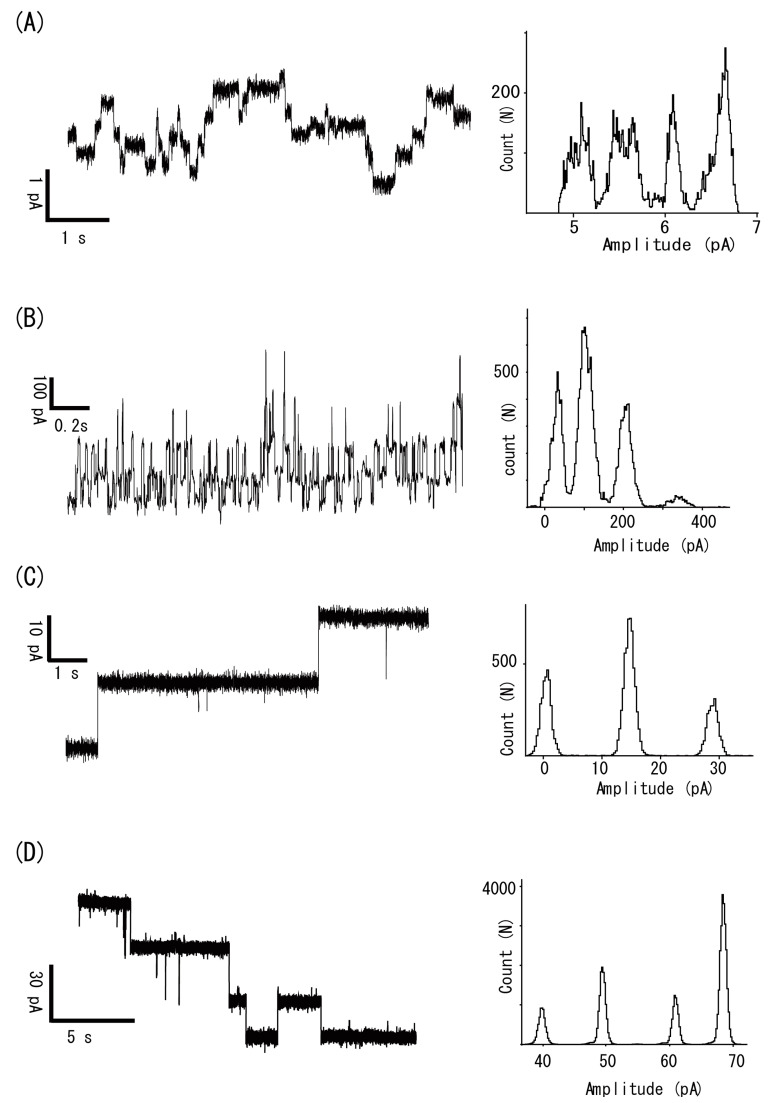
Channel current recordings via the spontaneous incorporation of peptides and proteins. (**A**) The gramicidin channel current was recorded with a recording solution containing 150 mM NaCl, 100 nM gramicidin, and 10 mM MOPS-Tris, pH 7.4. The trace was low-pass-filtered at 0.5 kHz. (**B**) The alamethicin channel current was recorded at 100 mV with a solution containing 1 M KCl and 500 nM alamethicin. (**C**) The α-hemolysin current was recorded at 60 mV with 100 mM KCl and 1 μM α-hemolysin, 10 mM Hepes-Tris, pH 7.3. (**D**) A channel current recording of mouse VDAC1. The recording solution contained 1 M KCl and 1 μg/mL VDAC1, 10 mM MOPS-Tris, pH 7.4. The recording was taken at 30 mV.

**Figure 5 micromachines-11-01070-f005:**
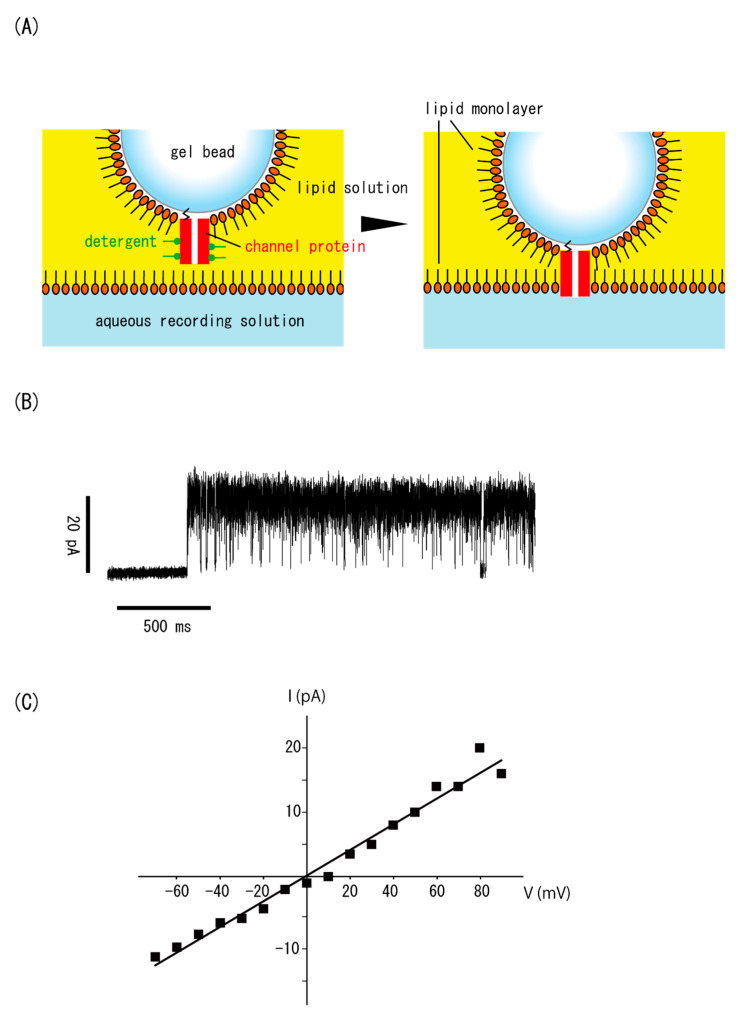
Recording of a single KcsA channel immobilized on a hydrogel bead. (**A**) Direct reconstitution of the KcsA channel into a bilayer. The KcsA channel protein was solubilized with detergent and immobilized on a gel bead via a histidine tag. The channel was incorporated into the bilayer immediately after the bilayer formation. (**B**) A typical current time course of the KcsA channel immobilized on a gel bead. (**C**) The single-channel current amplitude was plotted against the membrane voltage. From the slope of this line, the single-channel conductance was determined to be 200 pS.

**Figure 6 micromachines-11-01070-f006:**
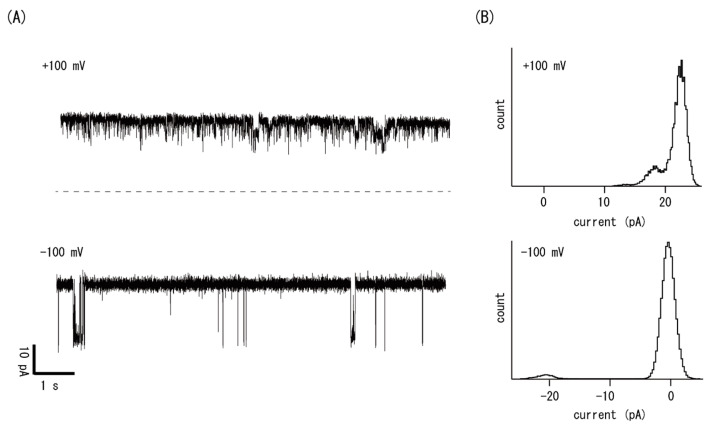
Single-channel current recordings of the voltage- and calcium-activated large conductance potassium channel (BK channel). (**A**) Current traces taken at indicated membrane voltages. (**B**) Current amplitude histograms at +100 mV and at −100 mV.

**Figure 7 micromachines-11-01070-f007:**
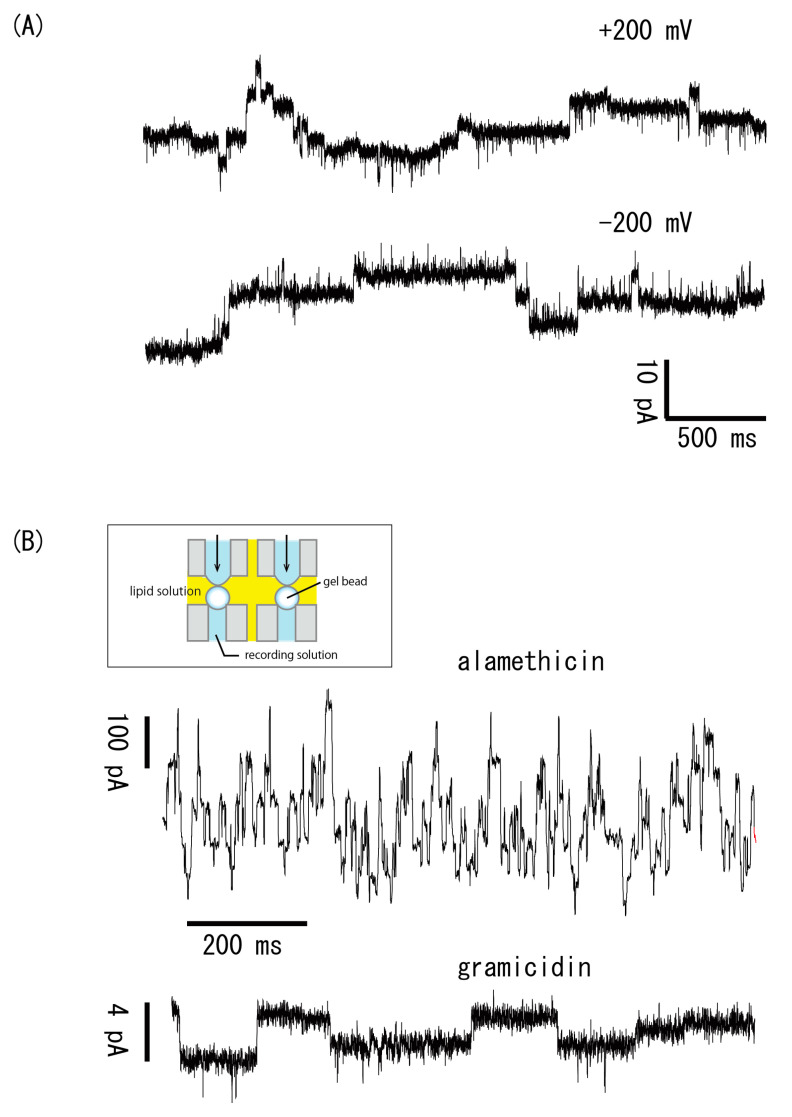
(**A**) Gramicidin currents recorded at ±200 mV. (**B**) Simultaneous recordings of gramicidin and alamethicin channels. The inset illustrates the recording apparatus.

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
