# Peer review of "A Lipid Bilayer Formed on a Hydrogel Bead for Single Ion Channel Recordings"

_micromachines, 2020, doi:10.3390/mi11121070_

Round 1

Reviewer 1 Report

This manuscript describes the use of hydrogel beads as the support for functionalised lipid bilayers, enabling single ion channel recordings.

As such, the work is an extension of the well-described droplet-interface bilayer system, and an extension of early black lipid membrane work.

While this is indeed an interesting research project, and the results look promising, the manuscript falls short in various aspects.

1) The introduction is very short, and in my view misses key background information. 

2) The methods section could be more explicit. Especially for this journal, and in a manuscript that introduced a new technological development, I would like to see all the information needed to easily reproduce the experiments. 

3) I don't fully understand chapter 3.1, mainly because I can't visualise how the experiment was conducted (see above). I would also like to see a complementary measurement validating the formation of a lipid bilayer. Would fluorescence microscopy work on your system?

4) In the single ion channel community, current traces are typically complemented by open/closed histograms. This should be added. 

5) I am missing information about reproducibility and stability. The current traces shown, how long is the raw data? What is the success rate? How reproducible are the results? 

Reviewer 2 Report

This manuscript presents a nice alternative solution to create stable lipid bilayer with facilitated protein reconstruction. The final goal will be miniaturization and automation for a potential better throughput in ion channel drug discovery. 

The senior author is well respected in the field and has a good overview on current development. Here the authors focused on gel-supported planar bilayer. I like their approach as it seems to combine the stability of a supprted membrane while maintaining the biocompatibility. Several channel forming proteins (of different type) has been characterized. 

This is a nice short report and I recommend publication. 

Author Response

We appreciate the time and effort you have dedicated to providing insightful feedback on ways to strengthen our paper. It is with great pleasure that we resubmit our article for further consideration.

Reviewer 3 Report

The article by Hirano et al. reports a method to form lipid bilayer incorporating ion channels on hydrogel beads for single ion-channel recordings. 

The work is an improvment on the methodological approach with respect to previous ones. The results are interesting and the capacity of this approach is validated on different proteins and also using different strategies of protein incorporation into the bilayer.

I find some concerns to be resolved before the manuscript may be accepted for publication. Namely:

1- In the methods section, it is not clear the lipid solution composition and preparation, and thus the final bilayer composition.

2- I think that the methodology of sample perapration should be improved. It is diffiult to follow for example the difference and purpose of those different steps between sections 2.6 and 2.6. As the most important report on the article is the methodology and verifications of it working, a very clear and anambiguous description of that should be described. This can be improved.

3- Is it possible to know or control the orientation of the channel proteins on the bilayer? can you include a coment on this aspect?

4- line 122, "figure" should read "figure 1A"

5- line 175, s¡you say it is not possible to confirm that the lipid bilayer was formed, only trace the channel recordings. That would be really helful, isn't there a way to check for the bilayer formation? ex-situ probably?

6- Figure 5C, the axis names and units are missing

7- section 3.4, how is the suspension of vesicles formed and composed (for the vesicle fusion process)?.

8- line 308-309, the phrase is missleading. What do you mean with "The bilayer, whose diameter was estimated by the microscopic image was approximately 10 um, ..."? is that the diameter of the gel bead? not the bilayer, right? MAybe it is necessary to rephrase.

9- the procedure of section 3.5 is not that clear to me...
